# *Zuccagnia punctata* Cav., a Potential Environmentally Friendly and Sustainable Bionematicide for the Control of Argentinean Horticultural Crops

**DOI:** 10.3390/plants12244104

**Published:** 2023-12-07

**Authors:** Sofía Manrique, Jessica Gómez, Mauricio Piñeiro, Belén Ariza Sampietro, Maria L. Peschiutta, Alejandro Tapia, Mario J. Simirgiotis, Beatriz Lima

**Affiliations:** 1Instituto de Biotecnología—Instituto de Ciencias Básicas, Universidad Nacional de San Juan (UNSJ), San Juan J5400ARL, Argentina; manriquesofia2@gmail.com (S.M.); jesicagomez674@gmail.com (J.G.); mauridpg@gmail.com (M.P.); barizasam88@gmail.com (B.A.S.); 2Consejo Nacional de Investigaciones Científicas y Técnicas (CONICET), CABA, Buenos Aires C1425FQB, Argentina; 3Instituto Multidisciplinario de Biología Vegetal (IMBIV-CONICET)—Cátedra de Química Orgánica, Facultad de Ciencias Exactas, Físicas y Naturales, Universidad Nacional de Córdoba, Córdoba X5000GYA, Argentina; mlaurapeschiutta@gmail.com; 4Instituto de Farmacia, Facultad de Ciencias, Campus Isla Teja, Universidad Austral de Chile, Valdivia 5090000, Chile; 5Center for Interdisciplinary Studies on the Nervous System (CISNe), Universidad Austral de Chile, Valdivia 5090000, Chile

**Keywords:** resin exudate, decoction, essential oil, *Meloidogyne incognita*, antioxidant

## Abstract

This research was designed to investigate the metabolite profiling, phenolics, and flavonoids content as well as the potential nematicidal properties of decoction (ZpDe), orange-yellow resin (ZpRe) and essential oil (ZpEO) from Argentinean medicinal plant *Zuccagnia punctata* Cav. Additionally, the antioxidant and antibacterial properties of ZpDe and ZpEO were determined. Metabolite profiling was obtained by an ultrahigh-resolution liquid chromatography MS analysis (UHPLC-ESI-QTOF/OT-MS-MS) and GCMS. The nematicidal activity was assayed by a standardized method against *Meloidogyne incognita*. The antioxidant properties were screened by four methods: (2,2-diphenyl-1-picrylhydrazyl assay (DPPH), Trolox equivalent antioxidant activity assay (TEAC), ferric-reducing antioxidant power assay (FRAP), and lipid peroxidation in erythrocytes (ILP). The antibacterial activity was evaluated according to the Clinical and Laboratory Standards Institute (CLSI) rules. The ZpDe, ZpRe and ZpEO displayed a strong nematicidal activity with an LC_50_ of 0.208, 0.017 and 0.142 mg/mL, respectively. On the other hand, the ZpDe showed a strong DPPH scavenging activity (IC_50_ = 28.54 µg/mL); ILP of 87.75% at 250 µg ZpDe/mL and moderated antimicrobial activity. The ZpEO showed promising activity against a panel of yeasts *Candida albicans* and non-albicans (ATCC and clinically isolated) with MIC values from 750 to 1500 µg/mL. The ZpDe showed a content of phenolics and flavonoid compounds of 241 mg GAE/g and 10 mg EQ/g, respectively. Fifty phenolic compounds were identified in ZpDe by ultrahigh-resolution liquid chromatography (UHPLC–PDA– Q-TOF-MS) analysis, while forty-six phenolic compounds were identified in ZpRe by UHPLC-ESI-Q-OT-MS-MS and twenty-nine in ZpEO using a GC-MS analysis, updating the knowledge on the chemical profile of this species. The results support and standardize this medicinal plant mainly as a potential environmentally friendly and sustainable bionematicide for the control of Argentinean horticultural crops including tomatoes and peppers and as a source of antimicrobial and antioxidant compounds which could be further explored and exploited for potential applications.

## 1. Introduction

The agricultural–livestock systems in Argentina face, in the present decade, a series of adversities due to economic, climatic and plant health factors, among others. In recent years, in some Andean areas of central-western Argentina, the quality and yields of horticultural products have suffered a decline, due to the persistent drought. The scarcity of snowfall in the Andes Mountains, which has a considerable impact on the flow of Andean rivers, does not allow adequate and sufficient artificial irrigation of crops, which, associated with the prevailing high temperatures, has contributed to the decrease in the quality and quantity of horticultural products.

In the Andean agricultural systems of central-western Argentina, another relevant factor that affects the yield of some cultivar are the pest species such as *Ceratitis capitata* (Wiedemann), the “Mediterranean fruit fly” (Diptera, Teprhitidae), which is a multivoltine and highly polyphagous species capable of utilizing at least 250 different hosts including fruits, nuts, and vegetables [1] including pears, apples, grapes and quinces causing mainly fruit losses [2,3]. Regarding horticultural cultivars such as tomatoes and peppers, a growing concern for farmers is the presence of plant parasitic nematodes (PPN) highlighting the presence of the genera of the genus Meloidogyne (Figure 1), which has been reported to produce up to 30% loss in crop yield [4]. Although there are more than 100 species [5], *M. incognita* has been the most recognized for a long time, because of its scientific and economic implications for the whole world [5,6]. This genus is known as root-knot nematode (RKN), due to the symptoms it generates in the root [7]. Root knots are the result of the feeding process and the habit of parasitism (endoparasite). The way in which they can ensure their nutrition throughout all their sedentary stages is by generating large cells, affecting the absorption functions of the roots [8].

Agrochemicals have had a major role in improving yields in food production. However, concerns have arisen about the negative impact that such chemicals have on human health and the environment [9,10,11]. Over the last decade, there has been an increasing search for new methods of controlling animal and plant parasitic nematodes, using methods less toxic to the environment and to people [3]. Among these, the search for compounds from natural sources stands out [12,13,14,15]. In this context, the search for new sources of potential nematicides from unexplored natural products, such as South American flora, has become relevant. The Andean plants represent a scarcely explored source that can offer promising candidates for the study of potential active extracts or new biomolecules of agronomic interest or incidence. The species that grow in these arid environments, some of them extreme, synthesize molecules and develop protection mechanisms, which allow them to adapt and survive in these ecosystems [16]. In this sense, glandular trichome on the surface leaves of resinous species stores substances to later secrete as resinous exudates, which allow them to retain moisture and function as a chemical barrier against the attack of phytoparasitic microorganisms and insects [16,17,18,19].

The Argentinean Andean region is the habitat of arid and semiarid land species such as *Bulnesia retama*, *Gochnatia glutinosa*, *Larrea nitida*, *Larrea divaricata* and *Zuccagnia punctata*, which are recognized for their high production of resins or exudates, of which there is a lack of knowledge about their potential as compounds of agronomic and industrial interest. Plants exudates are known to possess several biological activities including antimicrobial, antioxidant, anthelmintic, nematicidal among others [20,21]. However, a limited number of this Argentinean resinous species as a source of compounds of agronomic interest has been investigated until now.

*Zuccagnia punctata* is used in Argentina to treat injuries and bruises as a disinfectant for wounds, as a repellent of insects, for roof construction in rural areas and as a vegetable fuel for cooking food. In Argentina, *Z. punctata* is “the medicinal plant”: it has received the largest number of studies of its chemistry and biological activities, including antimicrobial, antioxidant, anti-mutagenic, anti-inflammatory, repellent and cytoprotective activities [19,22,23,24,25,26,27,28,29,30].

To our knowledge, there are no reports of properties of *Z. punctata* on plant parasitic nematodes (PPN). This work’s main goals and novelty are mainly the nematicidal effects against *M. incognita* of *Z. punctata* decoction (ZpDe), resin exudate (ZpRe), essential oil (ZpEO) and hydrosol (ZpEOH) to open a new eco-friendly and sustainable way of agronomic interest. Additionally, the antimicrobial and antioxidant effects are displayed and UHPLC-MS/GCMS chemical characterization of the ZpDe and ZpEO from *Z. punctata* collected in central-western Argentina is carried out.

## 2. Materials and Methods

### 2.1. Chemicals

Ultra-pure water (<5 µg/L TOC) was obtained from a water purification system Ari-um 126 61316-RO plus an Arium 611 UV unit (Sartorius, Goettingen, Germany). Methanol (HPLC grade) and formic acid (puriss. p.a. for mass spectrometry) were obtained from J. T. Baker (Phillipsburg, NJ, USA). Chloroform, ethyl acetate, methanol, chloroform, dichloromethane and hexane (HPLC grade) were obtained from Merck (Santiago, Chile). HPLC standards with purity higher than 95% were obtained from Sigma-Aldrich Chem. Co. (St. Louis, MO, USA) or Extrasynthèse (Genay, France).

### 2.2. Plant Material

The aerial parts of *Zuccagnia punctata* Cav. (Fabaceae, Caesalpinoideae) were collected in San Juan province, Argentina, at an altitude of 1000 m above sea level. A herbarium specimen previously identified by Dra. Gloria Barboza, IMBIV (Instituto Multidisciplinario de Biología Vegetal), Facultad de Ciencias Exactas, Físicas y Naturales, Universidad Nacional de Córdoba, Argentina, was deposited at the herbarium of the Botanic Museum of Córdoba (CORD 1125).

### 2.3. Extracts and Essential Oil

#### 2.3.1. *Z. punctata* Decoction (ZpDe)

Decoctions from *Z. punctata* aerial parts (ZpDe) were prepared at 10% weight/volume (associated or related to its popular or medicinal use), from 500 g of dried and milled plant (leaves), in 5 L. of purified water by means of PSA equipment. After 30 min of boiling, the decoctions were filtered, cooled for 24 h in a freezer at −40 °C, and then three representative samples of 100 mL of each decoction were subsequently lyophilized in an LA-B3 RIFICOR equipment (Buenos Aires, Argentina), obtaining a yield of 1.50 ± 0.01% *w*/*v*. The sample was stored in a freezer at −40 °C until its use in the nematicidal, antioxidant assays, phenolics and flavonoids quantification, as well as in UHPLC-ESI-QTOF-MS analysis according to [29].

#### 2.3.2. *Z. punctata* Orange-Yellow Resin (ZpRe)

The orange-yellow resin was obtained by dipping fresh aerial parts (500 g; 4 L of dichloromethane grade HPLC, 1 min), filtered and evaporated under reduced pressure to yield a semisolid yellow-orange resin (50 g, 10% yield *w*/*w*). The ZpRe was stored in a freezer at −40 °C until its use for bioassays, phenolics, and flavonoids identification/quantification as well as in ultrahigh-resolution liquid chromatography orbitrap MS analysis (UHPLC-ESI-OT-MS) analysis according to previous analysis [29]. ’ZpRe’s markers galangin, pinocembrin, 2′,4′-dihydroxychalcone and 2′,4′-dihydroxy-3′-methoxychalcone were obtained in our laboratory from ZpRe as previously described [22,25,26]. The purities of reference compounds were ≥98% as determined by HPLC–DAD.

The MeOH-soluble fraction of ZpRe (1.5 g) was applied to a Sephadex LH-20 column (column length, 40 cm; i.d., 5 cm), eluent methanol (MeOH). Some 21 fractions of 10 mL each were obtained. After TLC comparison (silica gel, hexane (H)/ethyl acetate (EtOAc)) 8:2 as the mobile phase, detection under UV light, and after spraying with diphenyl boric acid, ethanolamine complex in MeOH, fractions with similar TLC patterns were combined as follows: 1 (50 mg; fraction 1); 2 (20 mg, fraction 2); 3 (100 mg, fractions 3–4); 4 (78 mg, fractions 5), 5 (314 mg, fractions 6–7); 6 (180 mg, fractions 8), 7 (128 mg, fractions 9), 8 (172 mg, fractions 10), 9 (437 mg, fractions 11–14), 10 (19 mg, fractions 15–21).

A representative sample of bioactive fractions 5–8 was successively percolated by Sephadex LH-20 column (column length, 46 cm; diameter, 2 cm) equilibrated with H/MeOH/CHCl_3_(2:1:1) to afford 100 mg of pure 2′,4′-dihydroxy-3′-methoxychalcone and 80 mg of pure 2′,4′-dihydroxychalcone. The structures of both chalcones were confirmed by ^1^H and ^13^C NMR spectroscopy and HPLCMS/MS data, the results being in agreement with the literature data [22,25,26].

#### 2.3.3. Essential Oil Extraction and Chemical Analysis

Fresh aerial parts (500 g) were subjected to hydrodistillation for 2 h using a Clevenger-type apparatus. The yields were averaged over two experiments and calculated according to the dry weight of plant material. Essential oils (ZpEO) were stored at −20 °C in airtight micro-tubes prior to chemical analysis. Additionally, the hydrosol obtained (ZpEOH), also was preserved in a freezer until the nematicidal assay. Qualitative ZpEO data were determined by GC–FID and GC-MS. Gas chromatography-mass spectrometry analyses were carried out on a Hewlett-Packard 5890 II gas chromatograph coupled to a Hewlett-Packard 5989 B mass spectrometer, using a methyl silicone HP-5MS (crosslinked 5% PH ME Siloxane) capillary column (30 m × 0.25 mm), film thickness 0.25 µm. Samples were analyzed using the following GC-MS conditions: oven temperature program: 50–250 °C at 3 °C/min, carrier gas: helium, 1.5 mL/min; injection temperature: 250 °C, FID detector temperature: 300 °C; split mode ratio of 1:60. Additional parameters in the mass spectrometer unit: ion source temperature of 250 °C; ionizing voltage of 70 eV; scan range from *m*/*z* 35 to *m*/*z* 300. The identification of components was performed with the use of the volatile oil ADAMS library together with retention indices of reference compounds and built-in Wiley and NBS peak matching library search systems. Quantitative percentage composition was determined from the GC peak areas without correction factors [30].

### 2.4. Nematode Populations

*M. incognita* populations were collected from susceptible tomato cultivar (*Solanum lycopersicum*) from Médano de Oro district, Rawson, San Juan, Argentina. Subsequently, this population was reared on tomato seedlings in a pot in the laboratory. After 40 days, infected roots were collected and washed gently with tap water. Under a stereoscopic microscope at 1.6×, egg masses were hand-picked and put in 1% NaOCl solution for 4 min to dissolve the matrix gelatinous. They were incubated in a growth chamber at 28 ± 1 °C. After hatch, second-stage juveniles (J2) were collected up to 3 days old for the assays [31].

### 2.5. Nematicidal Assay

In glass Petri dishes of 50 mm diameter, thirty J2 were placed on 10 mL of the solution to be evaluated for LD, aqueous solutions were prepared at 0.05% *w*/*v* concentrations. For ZpDe; ZpRe and ZpEO, 50 mg were weighed and dissolved in 1 mL of MeOH, 0.3 mL of Tween-20 and filtered water to obtain a final volume of 100 mL. Fractions 5 and 8, aqueous solutions were prepared at 0.05% *w*/*v* concentration. As a negative control, a solution with the same solvent used in each treatment was prepared at the same concentration. As a positive control, Abamectin (Abamex^®^, San Diego, CA, USA) diluted in water was used at the concentration indicated by the manufacturer (0.0018% *v*/*v*). Five replicates per treatment were performed. Petri dishes were incubated in a chamber at 28 °C in the dark. Inactive individuals were counted every 24 h for 72 h. Inactive individuals were placed in water to confirm their mortality [32,33].

The statistical analysis of the data is detailed below: Inactive J2 counted at 24, 48 and 72 h were expressed in percentage and analyzed by ANOVA two ways (treatment, hours and interaction). Treatment means were compared using LSD Fisher. For mortality, J2 values were expressed in percentage. The Schneider–Orelli formula was applied to calculate the % correction for natural mortality in the negative control:MC% = (MT% − MCO%)/(100 − MCO%) × 100
where MC%: corrected mortality expressed as a percentage, MT%: treatment mortality expressed as a percentage, MCO%: negative control mortality expressed as a percent-age. These values were analyzed by ANOVA one way and treatments means were compared using LSD Fisher Data were analyzed using InfoStat (version 202l).

The lethal concentration 50 (LC_50_) and their confidence intervals were estimated by a sigmoid dose–response model using R Studio Version 1.4.1106 software library.

### 2.6. UHPLC Analysis of ZpDe and ZpRe

#### 2.6.1. Ultrahigh-Resolution Liquid Chromatography Analysis of ZpDe (UHPLC-ESI-QTOF-MS)

UHPLC-ESI-QTOF-MS Instrument; LC and MS Parameters: the separation and identification of secondary metabolites from ZpDe were carried out on a UHPLC-ESI-QTOF-MS system, equipped with UHPLC Ultimate 3000 RS with Chromeleon 6.8 software (Dionex GmbH, Idstein, Germany), and a Bruker maXis ESI-QTOF-MS. The chromatographic equipment consisted of a quaternary pump, an autosampler, a thermostated column compartment, and a photodiode array detector. The elution was performed using a binary gradient system with eluent (A) 0.1% formic acid in the water, eluent (B) 0.1% formic acid in acetonitrile, and the gradient: isocratic 1% B (0–2 min), 1–5% B (2–3 min), isocratic 5% B (3–5 min), 5–10% B (5–8 min), 10–30% B (8–30 min), 30–95% B (31–38 min), and 1% B isocratic (39–50 min). The separation was carried out with an acclaim Thermo 5 µm C18 80 Å (150 mm × 4.6 mm) column at a flow rate of 1.0 mL/min. ESI-QTOF-MS experiments in negative ion mode were recorded and the scanning range was between 100 and 1200 *m*/*z*. Electrospray ionization (ESI) conditions included capillary temperature of 200 °C, a capillary voltage of 2.0 kV, dry gas flow of 8 L/min, and a pressure of 2 bars for the nebulizer. The experiments were performed in automatic MS/MS mode. The structural characterization of the bioactive compounds was based on HR full MS, fragmentation patterns, and similarity with literature data. For the analysis, 5 mg of each extract was dissolved in 2 mL of methanol, passed through a polytetrafluoroethylene (PTFE) filter, and 10 µL was injected into the apparatus. MS data were analyzed using Bruker Data Analysis 4.0 (Bruker Daltonik GmbH, Bremen, Germany) and ACD lab spectrum processor (New York, NY, USA) software v2013 [29].

#### 2.6.2. UHPLC–PDA-OT-MS Analysis of the ZpRe

##### UHPLC–DAD–MS Instrument

An UHPLC-high-resolution MS machine (Thermo Dionex Ultimate 3000 system with PDA detector controlled by Chromeleon 7.3 software (Thermo Fisher Scientific, Waltham, MA, USA) hyphenated with a Thermo Q-Exactive MS focus (Thermo, Bremen, Germany) was used [16]. For the analysis, 5 mg of the resin was dissolved in 2 mL of methanol, filtered through a 200-µm PTFE (polytetrafluoroethylene)filter, and 10 µL was injected in the instrument, considering all specifications as reported [29].

##### LC Parameters and MS Parameters

Liquid chromatography was performed using an UHPLC C-18 column (150 × 4.6 mm Acclaim, ID, 2.5 µm; Thermo Fisher Scientific, Bremen, Germany) at 25 °C, hyphenated with a Thermo Q-Exactive MS focus (Thermo, Bremen, Germany). The detection wavelengths were 330,280, 254, and 354 nm, and photodiode array detectors were set from 200–800 nm. Solvent delivery was performed at 1 mL/min using ultra-pure water supplemented with 1% formic acid (A) and acetonitrile with 1% acid formic (B) and a program starting with 5% B at zero time, then maintenance at 5% B for 5 min, then a change to 30% B within 10 min, then maintenance at 30% B for 15 min, going to 70% B for 5 min, then maintaining 70% B for 10 min, and finally returning to 5% B in 10 min, keeping this condition for twelve additional minutes to achieve column stabilization before next injection of 20 µL. Standards and the resin dissolved in methanol were kept at 10 °C during storage in the autosampler. The HESI II and Orbitrap spectrometer parameters were optimized as previously reported [29], and as follows: Sheath gas flow rate, 75 units; auxiliary gas unit flow rate, 20; capillary temperature, 400 °C; auxiliary gas heater temperature, 500 °C; spray voltage, 2500 V (for ESI−); and S lens, RF level 30. Full scan data in positive and negative were acquired at a resolving power of 70,000 FWHM at *m*/*z* 200, even though negative mode is used for best performance. Scan range was from *m*/*z* 100 to 1000; automatic gain control (AGC) was set at 3 × 106 and the injection time was set to 200 ms. The chromatographic system was coupled to MS with a source II heated electro-nebulization ionization probe (HESI II). Nitrogen gas carrier (purity > 99.999%) was obtained from a Genius NM32LA (Peak Scientific, Billerica, MA, USA) generator and used as a collision and damping gas. The mass calibration for Orbitrap was performed every day, in order to ensure the accuracy of an operating mass equal to 5 ppm. A mixture of taurocholic acid sodium salt, buspirone hydrochloride, and sodium dodecyl sulfate (Sigma-Aldrich, Darmstadt, Germany), plus a fluorinated phosphazine solution, Ultramark 1621 (Alpha Aezar, Stevensville, MI, USA), was used as standard solution. All compounds were dissolved in a mixture of methanol, acetic acid, acetonitrile, water, (Merck, Santiago, Chile), and infused using a Chemyx Fusion 100 (Thermo Fisher Scientific, Bremen, Germany) syringe pump every day. The Q-Exactive 2.3 SP 2, XCalibur 2.4 and Trace Finder 3.3 (Thermo Fisher Scientific, Bremen, Germany) were used for UHPLC mass spectrometer control and data processing, respectively.

### 2.7. Determination of Total Phenolics and Flavonoids Content of ZeDe

The total phenolic and flavonoid content was determined by employing total phenols assay by Folin-Ciocalteu reagent and flavonoids by AlCl_3_ assay, both in microplate. The total phenolic was expressed as milligrams of gallic acid equivalents (GAE) per gram of extract (mg GAE/gZeDe). Flavonoids were expressed as milligrams of quercetin equivalents (QE) per gram of extracts (mg QE/gZeDe). The values were obtained using a Multiskan FC Microplate Photometer (Thermo Scientific, Waltham, MA, USA), and are shown as the mean ± standard deviation (SD) [29].

### 2.8. Antioxidant Activity

#### 2.8.1. Radical Scavenging Capacity Assay of 2,2-Diphenyl-1-picrylhydrazyl (DPPH)

The free radical scavenger effect of the ZpDe was assessed by the fade of a methanolic solution of the radical DPPH [29]. Extracts were assayed at concentrations between 1–1000 μg/mL. Scavenging activities were evaluated spectrophotometrically at 517 nm using the absorbance of the DPPH radical as a reference. The loss of color indicated the free radical scavenging efficiency of the substances. DPPH antioxidant capacity was calculated as follows:% scavenging effect = [1 − (Asample − Ablank)/ADPPH] × 100

The extract concentration providing 50% of radicals scavenging activity (EC_50_) was calculated from the graph plotting inhibition percentage at A_517_ against the extract concentration. Catechin (Sigma-Aldrich, ≥98%) was used as a reference compound (EC_50_ 4.1 μg/mL) [29].

#### 2.8.2. Ferric-Reducing Antioxidant Power Assay (FRAP)

The FRAP assay was run in microplate, as was the case with previously reported methodology [29]. Briefly, FRAP reagent and a methanolic solution of ZpDe (1 mg/mL) were mixed; simultaneously, a calibration curve was prepared by mixing FRAP reagent and Trolox solutions, at concentrations between 0 and 1 mmol/L. The absorbance values of mixtures were obtained in a Multiskan FC Microplate Photometer. Results were obtained by linear regression from the FRAP-Trolox calibration plot and show inequivalent milligrams Trolox/g ZpDe [29].

#### 2.8.3. Trolox Equivalent Antioxidant Activity Assay (TEAC)

TEAC assay was carried out in microplate following the previously reported methodology. Briefly, a ZpDe methanolic solution was mixed with 200 µL of ABTS, measuring their absorbance at 734 nm after 4 min. Results were obtained by linear regression from a calibration curve constructed with Trolox and are expressed as equivalent milligrams Trolox/g ZpDe [29].

#### 2.8.4. Inhibition of Lipid Peroxidation in Erythrocytes

The ability of the ZpDe at two concentrations (100 and 250 µgZpDe/mL) and of catechin at a single concentration (100 µg/mL) to inhibit lipoperoxidation in erythrocytes (LP), induced by tert-Butyl hydroperoxide, was determined. Relevant technical aspects of the trial have been reported recently in detail. The values obtained are expressed as percentages of lipid oxidation inhibition (ILP) [29].

### 2.9. Antibacterial Activity

#### 2.9.1. Microorganisms

For antibacterial evaluation, strains were used from the American Type Culture Collection (ATCC, Rockville, MD, USA) and clinical isolates from Laboratorio de Microbiología, Hospital Marcial Quiroga, San Juan, Argentina (MQ). The panel comprised the following bacteria: *Staphylococcus aureus* methicillin-sensitive ATCC 25923 (MSSA), *Staphylococcus aureus* methicillin-resistant ATCC 43300 (MRSA), *Staphylococcus aureus* methicillin-resistant-MQ1, *Staphylococcus aureus* methicillin-resistant-MQ2, *Salmonella* sp-MQ3 and *Escherichia coli* ATCC 25922. And also for the antifungal evaluation, we used strains from the American Type Culture Collection (ATCC), Rockville, MD, USA and CEREMIC (CCC), Reference Center in Mycology, Faculty of Biochemical and Pharmaceutical Sciences, Suipacha 531, 2000-Rosario, Argentina, and from Laboratorio de Microbiología, Hospital Marcial Quiroga, San Juan, Argentina (MQ). The following yeasts integrated the panel: *Candida albicans*-MQ 1924, *Candida glabrata* MQ1234, *Candida parapsilosis* MQ3 and *Candida tropicalis* CCC 131-2000.

#### 2.9.2. Antibacterial Susceptibility Testing

Minimum inhibitory concentration (MIC) of ZpDe and antibiotic Imipenem (commercial name Imipecil from Laboratory Northia, Buenos Aires, Argentina) was carried out by broth microdilution techniques, according to CLSI [34]. The ZpDe was tested from 3000 g/mL to 0.98 using an inoculum of each bacterium adjusted to 5 × 10^5^ cells with colony forming units (CFU)/mL. The absorbance at 620 nm was determined in a Multiskan FC Microplate Photometer (Thermo Scientific, Waltham, MA, USA). Also, the minimum bactericidal concentrations (MBCs) were determined, according to previous reports.

#### 2.9.3. Antifungal Susceptibility Testing

Minimum inhibitory concentration (MIC) of each extract or pure compound was determined by broth microdilution techniques, in according to the guidelines of the Clinical and Laboratory Standards Institute [35]. MIC values were determined in RPMI-1640 medium (Sigma, St. Louis, MO, USA), buffered to pH 7.0 with MOPS. Micro-titer trays were incubated at 35 °C for yeasts. MICs were visually recorded at 48 h for yeasts, and at a time according to the control fungus growth, for the rest of fungi. For the assay, stock solutions of ZpDe and ZpEO were two-fold diluted with RPMI medium from 3000 to 0.98 µg/mL (final volume 100 µL and final dimethyl sulfoxide (DMSO) concentration ≤ 1%). A volume of 100 µL of inoculum suspension was added to each well with the exception of the sterility control, where sterile water was instead added to the well. Ketoconazole (Sigma-Aldrich were used as positive control. End points were defined as the lowest concentration of drug resulting in total inhibition (MIC_100_) of visual growth compared to the growth in the control wells containing no antifungal. The minimum fungicidal concentration (MFC) of each extract was determined as follows: after determining the MIC, an aliquot of 5 µL was withdrawn from each clear well of the microtiter tray and plated onto a 150 mm RPMI-1640 agar plate buffered with MOPS (Remel, Lenexa, KS, USA). Inoculated plates were incubated at 30 °C, and MFCs were recorded after 48 h. The MFC was defined as the lowest concentration of each compound that resulted in total inhibition of visible growth in the plates. Inoculum of cell or spore suspensions was obtained according to reported procedures and adjusted to 1–5 × 10^3^ cells/spores with colony forming units (CFU)/mL [35].

### 2.10. Statistical Analysis

Data from different assays were analyzed using InfoStat (version 202l).

## 3. Results and Discussion

### 3.1. Chemical Composition

#### 3.1.1. UHPLC-q-TOF-ESI-MSn Analysis of ZeDe

Over the last decade, the chemical composition of *Z. punctata* has been updated through the report of new chemical compounds for the species, identified in extracts of both apolar and polar nature, obtained using solvents such as ethanol, methanol, and ethanol, and water, decoction and others. Studies applying technologies in tandem with UHPLC or mass-coupled GC have allowed the tentative identification of numerous phenolic compounds, flavonoids, chalcones and others. In this report, from MS spectra and MSn experiments data obtained by means of a UHPLC-ESI-QTOF-MS analysis in mode negative, fifteen compounds were tentatively identified in *Z. punctata* decoction, including phenolic acids, flavones, terpenes and fatty acids. The complete metabolome identification is shown below in Table 1 and Figure 2.

Some of the compounds were identified by spiking experiments with available standards (rhamnetin, rhamnacin and sakuranetin). From the fifteen compounds tentatively identified in the *Z. punctata* decoction, three correspond to phenolic acid (2, 3, 7); seven flavones (4, 6, 8, 9, 10, 11, 12), one chalcone (5), two terpenes (13, 16) and one fatty acid (14). Compound 5 (2′,4′-dihydroxy-3′-methoxychalcone) is one of the two most characteristic chalcones of this species, which has been previously isolated and identified in extracts obtained from the aerial parts, fruits and resin with ethanol, methanol and dichloromethane [22,25,26,27,28,29]. The main compound 2, with *m*/*z* 315.1699, was tentatively identified as geranyl caffeate (3,7-dimethyl-2,6-octadienyl caffeic acid ester); the presence of fragments at *m*/*z* 134 and 178, with ion at *m*/*z* 134 as the base peak, corresponding to the radical ions from the fragmentation of the caffeic acid moiety, support this assignation.

#### 3.1.2. UHPLC–PDA-OT-MS Analysis of the *Zuccagnia punctata* Resin (ZpRe)

In the exhaustive analysis of the dichloromethane extract from the *Z. punctata* resin (ZpRe), combining full mass spectra and MSn experiments, forty-six compounds were identified including flavones, chalcones, caffeic acid derivatives, coumaric acid esters, xanthene’s derivatives, trichocethenes, compounds of a phenolic nature and others. The analysis developed shows a general coincidence with a previous study recently published on the resin of this Argentine medicinal species collected in the center-west of Argentina, specifically in the province of San Juan. Several phenolic compounds from ZpRe were rapidly identified using available standards and previously isolated compounds were also identified by spectroscopic methods, including 2′,4′-dihydroxychalcone,2′,4′-dihydroxy-3′-methoxychalcone, galangin, pinocembrin and naringenin [19,22,26,29]. The molecular formula was obtained through high-resolution accurate mass analysis (HRAM) and matched the isotopic pattern. The complete metabolome identification is shown below in Appendix A.

#### 3.1.3. *Z. punctata* Essential Oil Composition (ZpEO)

The ZpEO showed a yield of 0.22% (*v*/*wt*.); and a δ25: 0.975 g/mL. Regarding the chemical profile, a total of 32 compounds constituents are reported in Table 2, highlighting the presence of alpha pinene (10.1%), mesitylene (10.51%), thuja-2,4(10)-diene (4.9%), p-cymene (18.6), limonene and 1,8-cineole(7.4%), linalool (7.5), 1,2,4-trimethylbenzene (5.9%), terpinene-4-olerpinene-4-ol (3.8%), para cymen-8-ol (3.4%) and alpha terpineol (1.5%).

### 3.2. Nematicidal Activity

#### 3.2.1. Nematicidal Activity of ZpDe

The ZpDe was tested in vitro against J2 *M. incognita* at 0.05% *w*/*v* in a range between 24–72 h. Additionally, the nematicidal activity and the lethal concentration 50 (LC_50_) were determined. The results are shown in Table 3. The decoction showed a potent nematostatic activity against the J2, between 24–48 h, without significant differences with the positive control (Abamectina, commercial name Fast 1.8^®^ from the company Gleba S.A., Buenos Aires, Argentina). On the other hand, after 72 h, the ZpDe showed a high nematicidal activity (95.24% ± 8.74), according to [32], without significant differences with Abamectin (98.89% ± 2.48). On the other hand, ZpDe showed a LC_50_ equal to 0.208% *w*/*v*.

#### 3.2.2. Nematicidal Activity of ZpRe, Bioactive Sephadex LH-20 Fraction 5-8 and Chalcones

The ZpRe at 0.05% *w*/*v* was evaluated in an in vitro assay against J2 *M. incognita* for 72 h. The results of ZpRe nematostatic activity and nematicidal activity are shown in Table 4. The ZpRe showed a potent nematostatic activity against the J2, between 24–48 h, without significant differences to the positive control (Abamectina). On the other hand, after 72 hs, the ZpRe showed a strong nematicidal activity (100 ± 0.00), according to [32], without significant differences with positive control Abamectin (98.89% ± 2.48). The ZpRe showed a LC_50_ equal to 0.017 mg/mL.

The ZpDe was fractionated and evaluated at 0.05% *w*/*v* in aqueous solution against J2 *M. incognita*. Through bioguided isolation by nematicidal activity assays, fractions 5–8 showed potent activity. From them, the main recognized chalcones 2′,4′-dihydroxy-3′-methoxychalcone and 2′,4′-dihydroxy-chalcone were isolated from fractions 5–8 by successive sephadex LH-20 columns. The use of Sephadex LH-20 eluted with the mixture of hexane, dichloromethane or chloroform and methanol in the proportion (2:1:1), has shown to be very efficient for the separation of phenolic metabolites with a difference of one methyl [22,25,36]. The two chalcones were evaluated in the nematicidal activity assay in a concentration range between 0.25–0.5 mg/mL, and in a period of time of 24–72 h. Both chalcones showed strong nematicidal activity greater than 91% for both concentrations during the entire period of time evaluated. No significant differences were observed between both chalcones.

The potent nematicidal activity of ZpRe could be associated, at least in part, with the presence of the majority of chalcones 2′,4′-dihydroxychalcone (29) and 2′,4′-dihydroxy-3′-methoxychalcone (30), which were identified and quantified by HPLC-DAD as characteristic biomarkers of this resinous Argentinean species in a previous report on samples collected in San Juan province, Argentina, together with other compounds. These are shown in Figure 3 [19,27]. The quantification of selected markers performed by the HPLC-UV method showed that the resin contains on average 3.18; 3.20; 16.04; and 12.84 g of pinocembrin (24), galangin (27), 2′,4′-dihydroxychalcone (29) and 2′,4′-dihydroxy-3′methoxychalcone (30), respectively, each quantified in 100 g of ZpRe [19]. Recently, the biological activities of the flavonoids 7-hydroxyflavanone, 3,7-dihydroxyflavone and 2′,4′-dihydroxychalcone from *Zuccagnia punctata* Cav. were examined in the free-living nematode *Caenorhabditis elegans*. From them, only 2′,4′-dihydroxychalcone showed an anthelmintic effect and alteration of egg hatching and larval development processes in *C. elegans*. Additionally, this chalcone was able to kill 50% of adult nematodes at a concentration of 17 μg/mL [37].

The J2 treated with ZpRe were observed under an optical microscope (magnification ×40) (Figure 4). The dead individuals adopted an upright position and an evident degree of internal vacuolization. This suggests that the osmotic condition could be altered in a manner similar to the intoxication effect described by [33]. The observed effect could be related to the V-ATPase enzyme complex, which is closely related to cuticle synthesis, osmotic regulation and detoxification of nematodes, which are essential processes in their life cycle. Some reports point to this enzyme complex as the site of action of many nematicides [38].

Plant-parasitic nematodes have caused huge economic losses to agriculture worldwide and seriously threaten the sustainable development of modern agriculture. Chemical nematicides are still the most effective means to manage nematodes. However, the long-term use of organophosphorus and carbamates nematicides has led to a lack of field control efficacy and increased nematode resistance. To meet the huge market demand and slow the growth of resistance, new nematicides need to enter the market [38].

The search for new alternative control strategies that are economically sustainable and environmentally sound at the same time for the protection of plants against phytoparasitic nematodes and soil-transmitted plant pathogens gives support to the exploration of the Andean flora that grows in Argentina, showing that some species, such as *Z. punctata*, are promising new sources for nematode control [39].

#### 3.2.3. Nematicidal Effect of ZpEO and Their Hydrosol ZpEOH

In Table 4, we can see the nematicidal activity of ZpEO against J2 *M. incognita* isolated from horticultural crops such as tomatoes and peppers growing in the central-western of Argentina. The ZpEO showed potent nematicidal activity comparable to Abamectin, according to the quantification ranges of nematicidal activity reported [32]. In previous reports on the chemical composition of *Zuccagnia punctata* oil collected in Argentine, including the province of San Juan, the presence of a major compound (-)-5,6-dehydrocrocamphor (18–62%) has been reported [30]. However, in this collected sample, (-)-5,6-dehydrocrocamphor was only detected in a low percentage, which supports the idea that it is in the presence of another chemotype of this species. The sample reported here also stands out for a high percentage of p-cymene (18%), limonene and 1,8-cineole (7.4%), as shown in Table 2.

Literature reports show that several essential oils and some of their main components, including terpinen-4-ol, linalool and alpha terpineol, have shown nematicidal activity against Meloidogyne and other important phytonematodes. Some terpenes that constitute oil support the potent nematicidal activity shown by the ZpEO. The EO of Kadsura heteroclite, contains as main components α-eudesmol (17.56%) and 4-terpineol (9.74%), and is nematicidal against *M. incognita* with an LC_50_ value of 122.94 μg/mL [40]. Some authors have demonstrated that oxygenated compounds including alcohols and ketones are generally more active than hydrocarbons. The essential oil of *Menta canadensis* aerial parts exhibited nematicidal activity against cereal cyst nematodes (Heteroderaavenae) and root-knot nematodes (Meloidogyne incognita) with LC_50_ values of 385.7 μg/mL and 139.0 μg/mL, respectively. The isolated constituents, menthol and α-terpinol, possessed nematicidal activity against *H. avenae* and *M. incognita* with LC_50_ values of 242.5 μg/mL, 190.3 μg/mL and 147.4 μg/mL, 115.2 μg/mL, respectively [41]. Geraniol, thymol, and terpinen-4-ol, also present in ZpEO, were shown to be moderately nematicidal, causing 100% paralysis of J2 exposed to 439, 1000, and 939 μL/mL after exposure of J2 for 24 h, and the EC_50_/24 h values were calculated at 237, 390, and 392 μL/mL, respectively, according to [42,43,44]. Sangwan and co-workers reported high activity of geraniol, eugenol, and linalool against *M. javanica* [45]. According to Walker and Merlin, these terpenes did not reduce irritations caused by *M. incognita* on tomato [46]. Some plant-derived nematicidal compounds including aldehydes such as furfural from Melia azedarach, (E,E)-2,4-decadienal and (E)-2-decenal from *Ailanthus altissima* and ketones such as 2-undecanone from *Ruta chalepensis* have been reported as having potential nematicidal activity in vitro and in pot experiments [39].

Regarding the hydrosol (ZpEOH), the nematicidal potential of ZpEOH was evaluated undiluted (ZpEOH_1_), diluted 1:2 (ZpEOH_2_) and diluted 1:4 (ZpEOH_3_) against J2 of *M. incognita*. The nematostatic activity of ZpEOH at the three concentrations tested was compared with that of the positive and negative control and the model was statistically significant (*p* < 0.0001). The effect of ZpEOH was dependent on the dilution, since it was observed that the ZpEOH_1_ produced total paralysis (100% ± 0.00) of the individuals in the time range of 24–72 h, while the ZpEOH_2_ and ZpEOH_3_ concentrations were ineffective for the same time range, after 72 h. It was confirmed that the paralyzed individuals were dead, and the efficiency of the treatment was calculated according to the Schneider–Orelli formula. The treatment efficiency data were analyzed statistically and the model was significant (*p* < 0.0001). It was observed that the nematicidal effect was positively correlated with the concentration used. The efficiency of the ZpEOH_1_ treatment was total (100% ± 0.00), according to the reference [32]. In turn, this treatment did not present significant differences to the positive control (98.96% ± 2.33).

The J2s treated with ZpEOH_1_ at 72 h were observed under an optical microscope (40× magnification). Figure 5A shows a treated specimen, seen from the entire body, in which a detachment of the cuticle and sectors of the atrophied soma can be observed. In Figure 5B,C, you can see these sectors in greater detail, which correspond to the anterior portion of the intestine, just before the junction with the basal bulb (Figure 5B) and in the final portion of the intestine, before the anal outlet (Figure 5C). Figure 5D shows an individual corresponding to the negative control at 72 h alive in which a conserved phenotype can be seen.

Hydrosols are a mixture formed by water (product of the condensation of the hydrodistillation process) and water-soluble compounds (carried away in the flow of the steam stream) such as acids, aldehydes or amines [47]. Therefore, the chemical composition of ZpEOH, which is polar in nature, differs from the composition of ZpEO, which is non-polar in nature. This is the first report of the nematicidal activity of ZpEOH. UHPLCMS analyses are currently being developed, with the objective of determining the chemical composition of ZpEOH_1_ and giving additional support to this byproduct as a sustainable alternative for the control of phytopathogenic nematodes.

Based on this, it could be inferred that the mechanism of action of ZpEOH_1_ could be different. A possible explanation could lie in the protein nature of the cuticle that covers nematodes [48]. Between the thiol and amino groups of the cuticle, nucleophilic addition reactions can occur with the carboxyl groups present in many organic acids (polar compounds). In this way, the cuticle could lose its waterproof function, leaving the internal organs exposed [49].

### 3.3. Total Phenolics and Flavonoids Contents and Antioxidant Activity of ZpDe

ZpDe was evaluated in vitro for total content of phenolics and flavonoids, in addition to antioxidant properties (Table 5). The extracts displayed a strong DPPH scavenging activity (EC_50_ = 28.54 ± 2.55 µg ZpDe/mL), as well as a good inhibition of lipoperoxidation in erythrocytes (87.75 ± 1.37 at 250 µg TLD/mL) comparable to the reference compound catechin (74% at 100 µg/mL). In respect to FRAP and ABTS antioxidants assays, the ZpDe showed a good effect in both trials, with 11.46 and 5.06 mg TE/mg of ZpDe, respectively. On the other hand, ZpDe showed a content of phenolic and flavonoid compounds of 241 mg GAE/g ZpDe and 10 mg QE/g ZpDe, respectively. The strong DPPH free radical scavenging activity is similar to the activity shown by the extract from the resin of this species. In a previous report, a strong free radical scavenging by ZpRe with an IC_50_ 25.72 µg/mL was reported as well as an good inhibition of lipid peroxidation in erythrocytes (70% percent at 100 µg ZpRe/mL) comparable to the reference compound catechin (74% at 100 µg/mL) [29]. In addition, the ZpRe showed a high content of TP, with values of 391 mg GAE/g ZpRe. From them, approximately eighty percent correspond to flavonoids (313 mg QE/g ZpRe). 

### 3.4. Antimicrobial Activity of ZpDe and ZpEO

Regarding ZpDe antimicrobial activity (Table 6), moderate activity is observed in bacteria with MIC values <3000 µg/mL, that activity being *Staphylococcus aureus* MQ2 (MIC = 2500 µg/mL), and *Salmonella* sp. (MIC = 2000 µg/mL), with respect to *E. coli* ATCC 25,922 (MIC = 3000 µg/mL). On the other hand, regarding yeasts, the extract did not show antifungal activity (MIC > 3000 µg/mL). The results of the ZpDe are similar to the MIC values shown by the decoction of other plants collected in the central west of Argentina, as is the case of the decoction of *Tessaria absinthiodes*, which showed a weak antibacterial activity, also following the CLSI guidelines, against Gram-positive bacteria, including *Staphylococcus aureus* methicillin-resistant ATCC 43300, *Staphylococcus aureus* methicillin-resistant-MQ-1, and *Staphylococcus aureus* methicillin-resistant-MQ-2 (MIC values between 2000 and 2500 µg/mL). Also, the decoction of *Tessaria absinthiodes* collected in Chile showed a weak activity against *Staphylococcus aureus* methicillin-resistant-MQ-2 (MIC = 2500 µg/mL) [50]. Also, these results are in accordance with a previous report on Argentinean medicinal plants to treat to bacterial infections [51]. Over the length of the Andes Mountains in South America (Peru, Ecuador, Chile, Bolivia and Argentina), numerous wild type plants are used as an alternative medical treatment. Because of this, these species are currently in high demand to treat infections caused by bacteria, filamentous fungi or yeasts and its marketing with herbalists and in popular markets is increasing. In general, the antimicrobial activity of the decoctions of Andean plants have shown higher values of MICs, with respect to ethanolic, methanolic, dichloromethane, and ethyl acetate extracts, among others. The synergistic mutual activity of *Zuccagnia punctata* Cav. (ZpRe) and *Larrea nitida* Cav. (LnRe) dichloromethane extracts (obtained by dipping) on clinical isolates of *Candida albicans* and *Candida glabrata* has been recently reported [19,27]. Also, in a previous report on *Azorella cryptantha*, another Andean medicinal plant, their extracts showed strong antibacterial activity against *S. enteritidis* with MIC values from 125 to 250 µg/mL, and towards methicillin-sensitive *Staphylococcus aureus*, and Gram-negative strains *E. coli*, *P. aeruginosa*, *Salmonella* sp. and *Yersinia enterocolítica*-PI, (MICs between 400 and 1000 g/mL) [52]. On the other hand, the ZpEO showed antimicrobial activity against typified bacteria and clinical isolates including *Staphylococcus aureus* methicillin-sensitive ATCC 25923, *Staphylococcus aureus* methicillin-resistant ATCC 43300, *S. aureus*- MQ1 and *Staphylococcus aureus* MQ2; *Escherichia coli* ATCC 25922 and *Salmonella* sp. with a minimum inhibitory concentration (MIC) between 1000 and 2000 μg/mL. The reference antibiotic Imipecil showed MICs in the range of 0.250 to 2 μg/mL. The ZpEO showed promising activity against a panel of yeasts of clinical incidence, which have been isolated at the Marcial Quiroga Hospital in San Juan, Argentina, including the following yeasts: *Candida albicans*-MQ 1924, *Candida glabrata* MQ1234, *Candida parapsilosis* MQ3 and *Candida tropicalis* CCC 131-2000, *C. tropicalis*. C131, *Cryptococcus neoformans* and *C. tropicalis* MQ1with MIC values from 750 to 1500 µg/mL. The MIC values obtained are within the activity ranges considered active by some authors. According Tamokou et al. [53], the extracts are very active if MIC values < 100 μg/mL, significantly active if 100 ≤ MIC ≤ 512 μg/mL, moderately active if 512 ≤ MIC ≤ 2048 μg/mL and not very active if MIC > 2048 μg/mL. In previous reports on ZpEO from aerial parts of nine populations of *Zuccagnia punctata* Cav. (Fabaceae) obtained by hydrodistillation and analyzed by GC–FID, GC–MS and ^13^C NMR, a total of 80 constituents, mainly oxygenated monoterpenes, were identified representing from 79.0 to 95.2% of the total oils which showed different composition patterns. The compound (-)-5,6-dehydrocamphor was the major constituent in six of the samples. The essential oil showed antifungal activity against the dermathophytes *Microsporum gypseum* and *Trichophyton mentagrophytes* with MIC values between 15.6 and 125 µg/mL, with *T. rubrum* being the most susceptible species [54]. Nowadays, approximately half of the reports of candidiasis are caused by non-albicans species, such as *C. glabrata*, *C. parapsilosis*, *C. tropicalis*, *C. krusei*, among others. The overall species distribution of Candida spp. Is dependent upon geographic location and patient population. The increase in C. parapsilosis in Latin America, southern European countries, and Africa, and *C. tropicalis* in Asia, has been reported [55,56,57]. The most frequently represented chemical constituents of EOs endowed with anti-Candida activity belong to the group of monoterpenes, and include *p*-cymene, linalool, γ-terpinene, carvacrol, 1-8-cineole, α-pinene, and thymol [58].

## 4. Conclusions

Plant-parasitic nematodes currently cause significant economic losses to agriculture worldwide, including horticultural crops developed in central western Argentina, seriously affecting the sustainable development of modern agriculture. The intensive use of organophosphate and carbamate nematicides has caused and is causing a decrease in the control of nematodes in horticultural crops, coupled with the development of increasing resistance of the different species of prevailing nematode populations. This has promoted the exploration of new sustainable sources of compounds or extracts with nematicidal potential. This paper reports, for the first time, the promising nematicidal activity against nematode J2 *M. incognita* of the resin extract (ZpRe), decoction (ZpDe), essential oil (ZpEO) and hydrosol (ZpEOH)of the medicinal species *Z. punctata*. A detailed chemical analysis of the different extracts is also reported, which provides support that the species is a source of compounds of different chemical structure and polarity. The 2′,4′-dihydroxychalcone and 2′,4′-dihydroxy-3′-methoxychalcone support, at least in part, the nematicidal activity shown by ZpRe. Currently, trials are being carried out in tomato pots with the aim of evaluating the nematicidal effect and how it affects the growth and development of the plants. The manuscript also makes an additional contribution to the antioxidant capacity of the decoction and the antimicrobial capacity of the essential oil against clinical isolates of *Candida* spp. as a promising source for human health.

## Figures and Tables

**Figure 1 plants-12-04104-f001:**
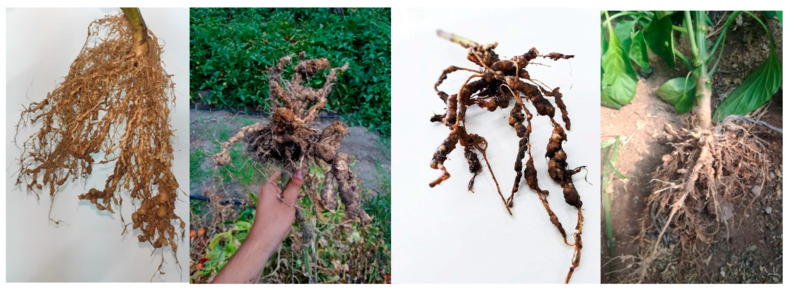
Roots of tomato (*Solanum lycopersicum*) and pepper (*Capsicum annuum*) plants with galls produced by a severe attack of *Meloidogyne incognita*.

**Figure 2 plants-12-04104-f002:**
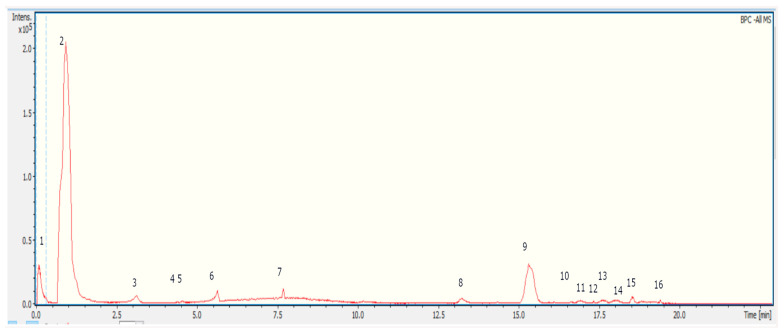
Representative UHPLC-MS (total ion current) chromatograms of ZpDe.

**Figure 3 plants-12-04104-f003:**
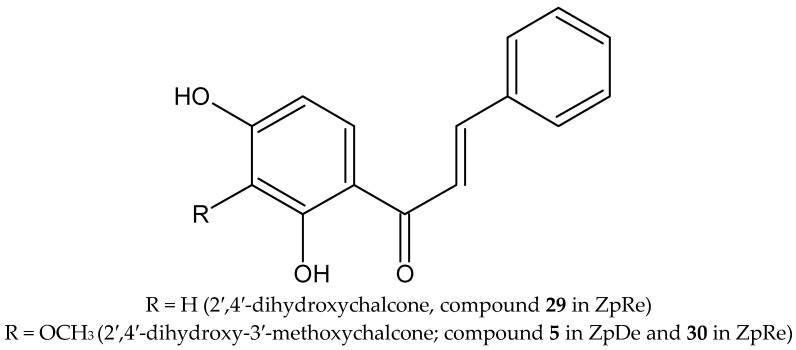
Chalcones with nematicidal activity from *Zuccagnia punctata* Cav.

**Figure 4 plants-12-04104-f004:**
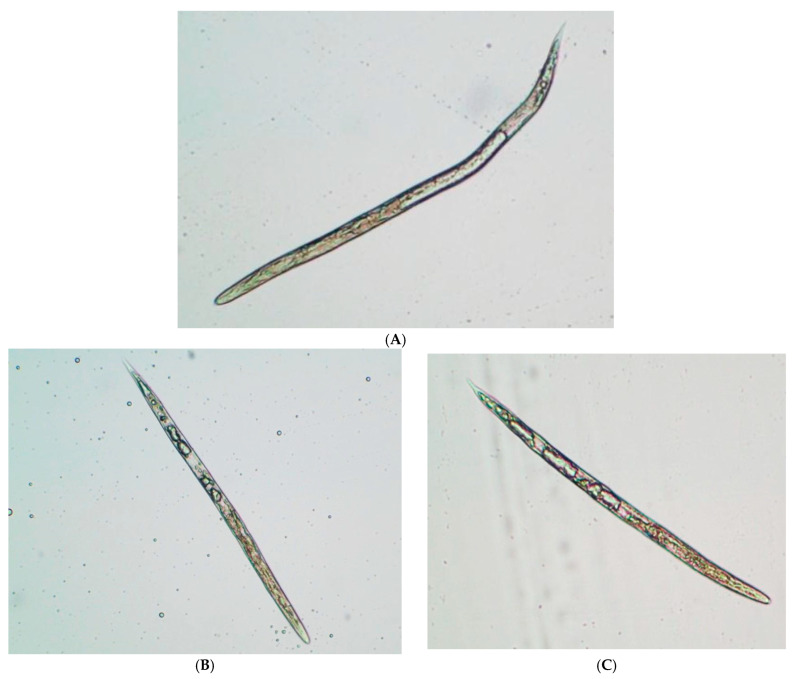
*M. incognita* J2 exposed at ZpRe after 72 h. (**A**) Negative control; (**B**) and (**C**) treated with ZpRe.

**Figure 5 plants-12-04104-f005:**
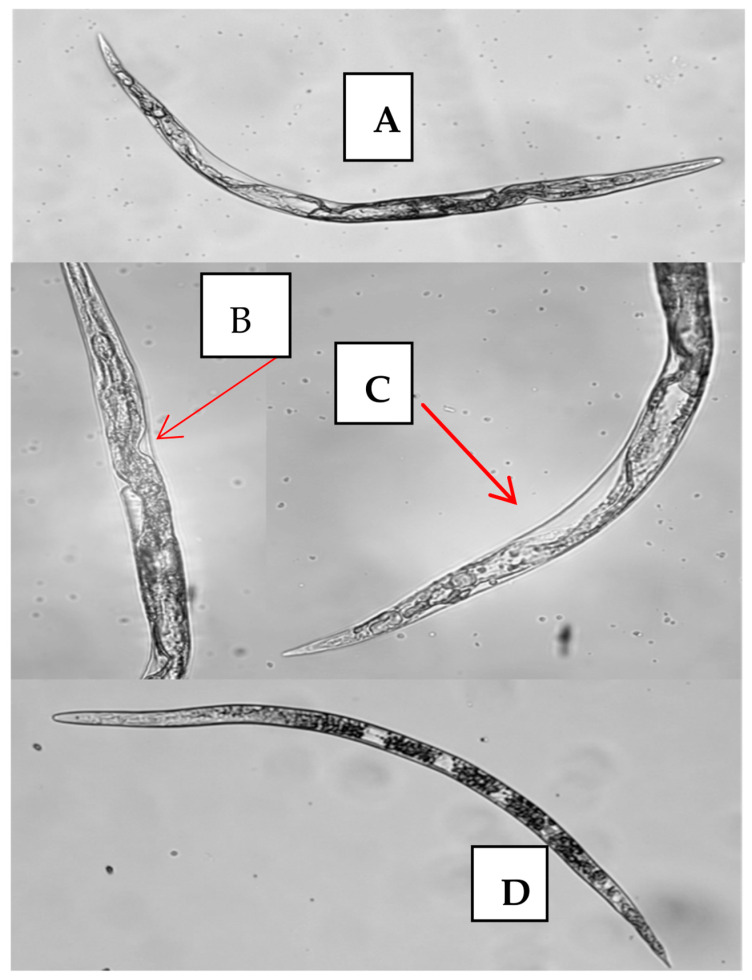
J2 of M. incognita: treated with ZpEOH_1_ after 72 h. (**A**) whole body (**B**) anterior region (**C**) posterior region (**D**) live negative control individual. The red arrows indicate the sectors where the epidermis separated from the cuticle. Observation under optical microscope 40× magnification.

**Table 1 plants-12-04104-t001:** High-resolution UHPLC—PDA—Q–TOF identification of metabolites from *Z. punctata* decoction (ZpDe).

Peak	Tentative Identification	[M-H]^−^	Retention Time (min.)	Theoretical Mass (*m*/*z*)	Measured Mass (*m*/*z*)	Accuracy (ppm)	Metabolite Type	MS Ions (ppm)
1	Na formiate (internal standard)	C_4_H_2_O_4_	0.22	112.9829	112.9856	3.1	Standard	588.8964, 656.8829, 724.8745
2	Geranyl caffeate	C_19_H_23_O_4_	1.22	315.1600	315.1699	−0.72	Phenolic acid	178.0265; 134.0364; 133. 0289
3	Rhapontin	C_21_H_23_O_9_	3.14	419.1347	419.1347	−0.0	Phenolic acid	271.05714, 269.0444, 255.06280
4	7,4′-dihydroxy-5-methoxy-flavanone	C_18_H_17_O_6_	4.52	285.07572	285.0763	−7.7	Flavone	149.9947; 119.0494
5	2′,4′-dihydroxy-3′-methoxychalcone	C_16_H_13_O_4_	4.76	269.08167	269.0808	−2.6	Chalcone	
6	Rhamnetin	C_16_H_11_O_7_	5.56	315.0511	315.0505	−2.6	Flavone	299.05616 (M-CH3), 279.1235, 255.0314
7	1-methyl-3-(3′,4′-dihydroxyphenyl)-propyl caffeic acid ester	C_19_H_19_O_6_	7.55	343.1187	343.1188	−3.19	Phenolic acid	135.0274
8	Rhamnacin	C_17_H_13_O_7_	13.23	329.0657	329.0666	−2.91	Flavone	315.0462, (M-CH3), 277.1075, 300.05554, 151.0020, 256.03405
9	3,7-dihydroxyflavone	C_15_H_9_O_4_	15.55	253.05025	253.0495	2.94	Flavone	208.0522; 223.0324; 195.0455; 180.0565
10	Sakuranetin	C_16_H_13_O_5_	16.57	285.0768	285.0759	−2.96	Flavone	247.0972
11	Eupatorin	C_18_H_15_O_7_	17.54	343.0823	343.0823	0.03	Flavone	321.1702
12	Circimaritin	C_17_H_13_O_6_	17.67	313.07412	313.07176	7.5	Flavone	165.02860, 239.05757
13	Ganoderiol C	C_32_H_53_O_5_	17.89	517.3898	517.3913	−2.83	Terpene	467.44415, 283.2615
14	Tetracosanoic acid	C_24_H_48_O_2_	18.01	367.3599	367.3581	4.79	Fatty acid	311.1610
15	Shinflavanone	C_25_H_25_O_4_	18.54	389.1765	389.1758	1.94	Flavone	289.0869
16	Neochlorogenin	C_27_H_43_O_4_	19.56	431.31739	431.31668	1.64	Terpene	863.6363, 344.13611, 279.23113, 321.2050

**Table 2 plants-12-04104-t002:** GC-MS analysis of ZpEO.

Peak	Compounds	RI	Area%	Identification Method
1	alpha pinene	938	10.1	1, 2
2	alpha fenchene	950	0.3	1
3	thuja-2,4-(10)-diene	960	4.9	1
4	beta pinene	980	0.2	1, 2
5	mesitylene	998	10.5	1
6	ethyl hexanoate	1000	0.3	1
7	alpha phellandrene	1003	0.5	1, 2
8	p-methyl anisole	1013	2.0	1
9	1,2,4-trimethylbenzene	1022	5.9	1
10	p-cymene	1025	18.6	1, 2
11	limonene	1028	7.4	1, 2
12	1,8-cineole	1031	7.4	1, 2
13	1,3-Cyclopentadiene, 5,5-dimethyl-1-ethyl-	1040	0.2	1
14	trans-Linalool Oxide	1089	0.8	1
15	p- cymenene	1091	1.9	1
16	1,7,7-Trimethylbicyclo [2.2.1]hept-5-en-2-one (-)-5,6-dehydrocrocamphor	1094	1.5	1
17	linalool	1097	7.5	1, 2
18	mentha-2,8-dien-1-ol	1133	1.3	1
19	cisverbenol	1139	1.3	1
20	p-mentha-1,5-dien-8-ol (alpha phellandren-8-ol)	1170	1.6	1
21	terpinen-4-ol	1178	3.8	1, 2
22	para cymen-8-ol	1181	3.4	1
23	alpha terpineol	1188	1.5	1, 2
24	2-allyl-phenol	1193	1.5	1
25	verbenone	1205	1.5	1, 2
26	para cymen-9-ol	1205	1.6	1
27	2-E-1-propenyl phenol	1263	0.4	1
28	2,3,6-trimethylbenzaldehyde	1355	0.9	1
29	arbozol (endo)	1434	0.1	1
30	arbozol (exo)	1454	0.1	1
31	delta cadinene	1520	0.4	1
32	beta eudesmol	1651	0.6	1
*Total*			100.0	

Constituents listed in order of increasing retention indices (RI). Unidentified components less than 0.1% are not reported. Temperature-programmed RI referred on-alkanes, determined on a HP-5MS capillary column. Method of identification of minor constituents: 1 corresponds to comparison of GC-MS data and RI with those of the volatile oil ADAMS Wiley and NBS computer mass libraries, 2 corresponds to comparison of GC-MS data and RI with those of authentic samples.

**Table 3 plants-12-04104-t003:** Nematicidal activity of ZpDe against J2 M. incognita.

Treatment	Nemostatic Activity			Nematicidal Activity (%)	LC_50_(mg/mL) (I.L-S-L)
	24 h	48 h	72 h		
ZpDe	66.43 ^b^ ± 20.6	89.44 ^bc^ ± 13.6	92.50 ^bc^ ± 11.7	95.24 ^a^ ± 8.74	0.208 (0.159–0.258)
Positive control	95.00 ^c^ ± 5.0	98.00 ^c^ ± 2.74	99.00 ^c^ ± 2.24	98.89 ^a^ ± 2.48	
Negative control	1.25 ^a^ ± 2.5	8.75 ^a^ ± 2.5	10.00 ^a^ ± 0.0		

Different letters between columns indicate a significant difference (*p* < 0.05) according to LSD Fisher.

**Table 4 plants-12-04104-t004:** Nematostatic activity, nematicide activity and LC_50_, ZpRe and fractions (5–8) and ZpEO against J2 *M. incognita*.

Treatment	Nemostatic Activity			Nematicide Activity (%)	LC_50_ (mg/mL) (I.L–S-L)
	24 h	48 h	72 h		
ZpRe	45.00 ^b^ ± 3.54	95.83 ^b^ ± 10.21	100 ^b^ ± 0.00	100 ± 0.00	0.017 (0.012–0.021)
Positive control	95.00 ^c^ ± 5.0	98.00 ^b^ ± 2.74	99.00 ^c^ ± 2.24	98.89 ± 2.26	
Negative control	0.40 ^a^ ± 0.89	1.40 ^a^ ± 2.19	1.40 ^a^ ± 2.19		
F(5–8)	95.0 ^c^ ± 5.0	96.67 ^c^ ± 2.89	96.67 ^c^ ± 2.89	96.60 ^a^ % ± 2.95	0.003 (0.002–0.005)
Positive control	92. 0 ^c^ ± 5.7	93.0 ^c^ ± 5.7	97.0 ^c^ ± 4.47	96.64 ^a^ % ± 4.56	
Negative control	0.0 ^a^ ± 0.0	0.0 ^a^ ± 0.0	2.0 ^a^ ± 2.74		
ZpEO	96.0 ^b^ ± 5.48	98.0 ^b^ ± 4.47	99.0 ^b^ ± 2.24	98.99 ^a^ ± 2.27	0.142 (0.101–0.183)
Positive control	95. 0 ^b^ ± 5.0	98.0 ^b^ ± 2.74	99.0 ^b^ ± 2.24	96.64 ^a^ ± 4.56	
Negative control	0.0 ^a^ ± 0.0	0.4 ^a^ ± 0.89	1.4 ^a^ ± 2.19		

Different letters between same columns indicate significant difference (*p* < 0.05).

**Table 5 plants-12-04104-t005:** Total phenolics and flavonoids content and antioxidant assays of ZpDe.

Assays	ZpDe
Phenolic compounds	241.34 ± 15.93
Flavonoids (mg QE/gZpDe)	10.03 ± 1.25
Antioxidant	
DPPH (EC_50_ in µgZpDe/mL)	28.54 ± 2.55
FRAP (mg TE/mg of ZpDe)	11.46 ± 0.16
TEAC (mg TE/g of ZpDe)	5.05 ± 0.01
ILP at 250 µgZpDe/mL	87.75 ± 1.37

**Table 6 plants-12-04104-t006:** Antimicrobial activity of ZpDe and ZpEO (minimum inhibitory concentrations (MICs), minimum bactericidal concentrations (MBCs) and minimum fungicidal concentrations (MFCs) in µg ZpDe/mL.

Microorganisms	Extracts				Reference Antibiotics	
Bacterias	ZpDe		ZpEO		Imipecil	
Gram (+)	MIC	MBC	MIC	MBC	MIC	MBC
MSSA	>3000	>3000	>3000	-	0.25	0.25
MRSA	3000	3000	>3000	-	1	1
*Staphylococcus aureus* MQ1	3000	>3000	>3000	-	2	2
*Staphylococcus aureus* MQ2	2500	3000	>3000	-	0.25	0.25
Gram (−)						
*Escherichia coli* ATCC 25922	3000	3000	>3000	-	0.62	0.62
*Salmonella* sp.	2000	>3000	>3000	-	0.5	1
Fungi					Ketoconazole	
	MIC	MFC	MIC	MFC	MIC	MFC
*C. albicans* MQ-1924	>3000	-	750	>2000	0.62	1.25
*C. glabrata* MQ-1	>3000	-	1500	>2000	2.5	>2.5
*C. tropicalis* MQ-C131	>3000	-	750	>2000	0.31	0.62
*C. parapsilopsis* MQ-1	>3000	-	750	>2000	0.15	0.15
*Cryptococcus neoformans* MQ-1	>3000	-	750	1000	0.62	2.5
*C. tropicalis* MQ1	>3000	-	750	>2000	0.62	0.62

MSSA: Methicillin-sensitive *Staphylococcus aureus* ATCC 25923; MRSA: Methicillin-resistant. *S. aureus* ATCC 43300; MIC: Minimum inhibitory concentration, MBC: Minimum Bactericidal Concentration; and MFC: Minimum Fungicidal Concentration in µg/mL.

## Data Availability

Data are contained within the article and Appendix A.

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
