# Peer review of "Zuccagnia punctata Cav., a Potential Environmentally Friendly and Sustainable Bionematicide for the Control of Argentinean Horticultural Crops"

_plants, 2023, doi:10.3390/plants12244104_

Round 1

Reviewer 1 Report

Comments and Suggestions for Authors

Manuscript titled "Zuccagnia punctata Cav., a potential environmentally friendly and sustainable bionematicide for the control of Argentinean horticultural crops’’. The study reported the chemical composition (32 compounds) of Zuccagnia punctata Cav EO extracted by hydro-distillation, as well as the identification of 15 polar compounds of metabolites from Z. punctata decoction by High-resolution UHPLC-PDA-Q-TOF identification, as well as the identification of 46 polar compounds, primarily Flavonoids. The EO, water, and DCM extracts were tested for nematicidal, antibacterial, and antioxidant activities. The manuscript is

interesting and finally provides useful information to readers; nonetheless, the authors must address a number of concerns.

- Compound 2 is naringenin, whereas compound 7 is the enantiomer of naringenin. Enantiomers cannot be separated without the use of a chiral column. Check your finding.

- Could you clarify how you were able to identify polar chemicals using only a few standards?

- The chemical structures of the key chemicals, particularly chalcones and flavonoids, must be included.

-Line 40: Fifty phenolics compounds were found in ZpDe by ultrahigh resolution liquid chromatography (UHPLC-PDA-Q-TOF-MS) analysis, however in Line 430 you stated that only 15 compounds were detected and in Table 1 16 compounds were identified.

-In Table 2, you indicated that you identified 32 components from the chemical composition of EO, accounting for 100% of the total area. Line 485, on the other hand, states that unidentified components less than 0.1% are not recorded. This signifies that the compounds identified accounted for less than 100% of the total area.

Comments on the Quality of English Language

good

Author Response

         Answers to Reviewer 1. Comments and Suggestions for Authors

Dear Reviewer. We appreciate and thank you for your comments, which have been considered, responded to point by point and incorporated into the version revised of manuscript with tracked changes and highlighted in yellow. Our manuscript has improved its presentation based on them.

Manuscript titled "Zuccagnia punctata Cav., a potential environmentally friendly and sustainable bionematicide for the control of Argentinean horticultural crops’’. The study reported the chemical composition (32 compounds) of Zuccagnia punctata Cav EO extracted by hydro-distillation, as well as the identification of 15 polar compounds of metabolites from Z. punctata decoction by High-resolution UHPLC-PDA-Q-TOF identification, as well as the identification of 46 polar compounds, primarily Flavonoids. The EO, water, and DCM extracts were tested for nematicidal, antibacterial, and antioxidant activities.

The manuscript is interesting and finally provides useful information to readers; nonetheless, the authors must address a number of concerns.

1- Compound 2 is naringenin, whereas compound 7 is the enantiomer of naringenin. Enantiomers cannot be separated without the use of a chiral column. Check your finding.

We agree with your observation, sorry for our mistake, where it says enatiomer it should say isomer (could be butein)

 The compound 7 Naringenin enatiomer was changed by Naringenin isomer (could be butein) In Suplementary Material Table S2

2- Could you clarify how you were able to identify polar chemicals using only a few standards?

The medicinal species Z. punctata is one of the Argentine species, on which numerous publications have been made on different biological activities and chemical updating of its constituents, including isolation, identification by GC-MS, UHPLCMS (QTOF, OT and others). This makes it possible to have data on compounds (phenolics, flavonoids and others) obtained through exhaustive UHPLCMS analysis, which allows rapid and unequivocal identification of several of its constituents, reported here. The reverse phase RP 18 column is able to separate polar compounds with polar solvent systems, which could be easily identified by the use of MS spectrometer or together with UV-vis absorbance.

Additionally, an access strategy to several databases (MONA mass spectrometry, metaboscape, etc) and through the use of specific software BRUKER DATA ANALYSIS and ACD lab spectrums processor, has allowed us to chemically update this species and others, which we have recently reported.

3- The chemical structures of the key chemicals, particularly chalcones and flavonoids, must be included.

The structure of chalcones with nematicidal activity from Zuccagnia punctata have been incorporated into a figure 3 as suggested

4-Line 40: Fifty phenolics compounds were found in ZpDe by ultrahigh resolution liquid chromatography (UHPLC-PDA-Q-TOF-MS) analysis, however in Line 430 you stated that only 15 compounds were detected and in Table 1 16 compounds were identified.

 In ZpDe were characterized only fifty phenolics compounds (2-16).

Compound 1 in Table 1 corresponds to the Na formatiate compound (internal standard), which is used for the equipment calibration process. Therefore, 16 compounds are reported in Table 1.

Table 1. High-resolution UHPLC–PDA– Q-TOF identification of metabolites from Z. punctata decoction (ZpDe)

5- In Table 2, you indicated that you identified 32 components from the chemical composition of EO, accounting for 100% of the total area. Line 485, on the other hand, states that unidentified components less than 0.1% are not recorded. This signifies that the compounds identified accounted for less than 100% of the total area.

The footnote of the Table 2 has been corrected. The paragraph “Percentage values less than 0.1% are denoted as (traces)” was deleted, we apologize for the mistake

Reviewer 2 Report

Comments and Suggestions for Authors

Congratulations on the idea. A very good approach to plant protection. The use of new solutions in plant protection, bionematocides, is currently the most important task of modern plant protection. Good luck.

Comments on the Quality of English Language

no comment

Author Response

Answers to Reviewer 2. Comments and Suggestions for Authors

Dear Reviewer. We appreciate and thank you for your comments, which have been considered and incorporated into the version revised of manuscript with tracked changes and highlighted in yellow. Our manuscript has improved its presentation based on them.

  • Keywords: Meloidogyne incognita was changed by Meloidogyne incognita
  • 3.1. Z. punctata decoction (ZpDe)

Decoctions from Z punctata aerial parts (ZpDe) was prepared at 10% weight/volume …

The decoction has been prepared at four concentrations, different times and with and without sonication, in a matrix generated for the optimization of the extraction process of bioactive compounds (antibacterial, nematicide and antifungal) from Zuccagnia punctata collected at different altitudes in the province of San Juan.

Here we are only reporting one concentration (10%), which is a concentration associated or related to its popular or medicinal use.

The paragraph:” Decoctions from Z punctata aerial parts (ZpDe) was prepared at 10% weight/volume, from 500 g of dried and milled plant (leaves), in 5 L. of purified water by means of a PSA equipment.”

Was changed by

“Decoctions from Z punctata aerial parts (ZpDe) was prepared at 10% weight/volume (associated or related to its popular or medicinal use), from 500 g of dried and milled plant (leaves), in 5 L. of purified water by means of a PSA equipment.”

Reviewer 3 Report

Comments and Suggestions for Authors

  The work "Zuccagnia punctata Cav., a potentially environmentally friendly and sustainable bionematicides for the control of Argentinean horticultural crops'' presents a very interesting and particularly elaborate study on the potential nematicidal properties of the decoction (ZpDe), the yellow-orange resin (ZpRe) and the essential oil (ZpEO) from the Argentinian medicinal plant Zuccagnia punctata Cav.

The study highlights the potential of the plant as an ecological and sustainable bionematicide for the control of horticultural crops, but also as a source of antimicrobial and antioxidant compounds, as an alternative to organophosphate and carbamate nematicides that lead to the development of resistance over time.

I recommend publishing the work after a minor revision.

1. I would suggest additional proofreading to check for grammatical errors and typos

2. I recommend the authors check the numbering of some subchapters

- the numbering of subchapter 2.3.1 should be corrected. Z. punctata Orange-Yellow Resin (ZpRe) as 2.3.2

- 2.6.2. 2 UHPLC–DAD–MS Instrument should be 2.6.2.1

-2.6.2.3. LC Parameters and MS Parameters should be 2.6.2.2

-3.2.2. UHPLC–PDA-OT-MS Analysis of the Zuccagnia punctata resin (ZpRe) should be 3.1.2

- 3.2.3 Z. punctata Essential Oil Composition (ZpEO) should be 3.1.3

- Chapter 3.2 is missing or the numbering is still wrong and must be corrected accordingly

- On row 448, the number of the figure must be corrected

- On line 574, the number of the figure must be corrected

- On row 665, the number of the figure must be corrected

3. I suggest the authors to emphasize the novelty of the study

4. More recent references on the subject could be brought.

Comments on the Quality of English Language

Minor editing of English language required.

Author Response

Answers to Reviewer 3. Comments and Suggestions for Authors

Dear Reviewer. We appreciate and thank you for your comments, which have been considered, responded to point by point and incorporated into the version revised of manuscript with tracked changes and highlighted in yellow. Our manuscript has improved its presentation based on them.

The work "Zuccagnia punctata Cav., a potentially environmentally friendly and sustainable bionematicides for the control of Argentinean horticultural crops'' presents a very interesting and particularly elaborate study on the potential nematicidal properties of the decoction (ZpDe), the yellow-orange resin (ZpRe) and the essential oil (ZpEO) from the Argentinian medicinal plant Zuccagnia punctata Cav.

The study highlights the potential of the plant as an ecological and sustainable bionematicide for the control of horticultural crops, but also as a source of antimicrobial and antioxidant compounds, as an alternative to organophosphate and carbamate nematicides that lead to the development of resistance over time.

I recommend publishing the work after a minor revision.

  1. I would suggest additional proofreading to check for grammatical errors and typos

A additional proofreading was carried out. Several

  1. I recommend the authors check the numbering of some subchapters

Several punctuation and spacing errors, due to errors in our computer systems during the writing and writing process, have been highlighted with tracked changes.

  • the numbering of subchapter 2.3.1 should be corrected. Z. punctata Orange-Yellow Resin (ZpRe) as 2.3.2

Was changed

  • Section 2.6.2. 2 UHPLC–DAD–MS Instrument should be 2.6.2.1

Was changed

  • Section-2.6.2.3. LC Parameters and MS Parameters should be 2.6.2.2

            Was changed

  • Section -3.2.2. UHPLC–PDA-OT-MS Analysis of the Zuccagnia punctata resin (ZpRe) should be 3.1.2

           Was changed

  • Section - 3.2.3 Z. punctata Essential Oil Composition (ZpEO) should be 3.1.3

           Was changed

  • - Chapter 3.2 is missing or the numbering is still wrong and must be corrected accordingly

         Was revised and corrected

  • - On row 448, the number of the figure must be corrected

           Was revised and changed

  • - On line 574, the number of the figure must be corrected

                 Was revised and changed

  • - On row 665, the number of the figure must be corrected

              Was revised and changed

  • I suggest the authors to emphasize the novelty of the study

          The conclusion was improved

  • More recent references on the subject could be brought.

Four references were included

The following paragraph was included

Nowaday, approximately half of the reports of candidiasis are caused by non-albicans species, such as C. glabrata, C. parapsilosis, C. tropicalis, C. krusei, among others. The overall species distribution of Candida spp. is dependent upon geographic location and patient population. The increase of C. parapsilosis in Latin America, southern European countries, and Africa, and C. tropicalis in Asia has been reported [55-57]. The most frequently represented chemical constituents of EOs endowed with anti-Candida activity belong to the group of monoterpenes, and include p-cymene, linalool, -terpinene, carvacrol, 1-8-cineole, -pinene, and thymol [58].

[55]. Lamoth, F.; Lockhart, S.R.; Berkow, E.L.; Calandra, T. Changes in the epidemiological landscape of invasive candidiasis. J. Antimicrob. Chemother. 2018, 73, i4–i13.

[56]. Ghazi, S.; Rafei, R.; Osman, M.; El Safadi, D.; Mallat, H.; Papon, N.; Dabboussi, F.; Bouchara, J.-P.; Hamze, M.The epidemiology of Candida species in the Middle East and North Africa. J. Mycol. Med. 2019, 29, 245–252.

[57]. Colombo, A.L.; de Almeida Júnior, J.N.; Guinea, J. Emerging multidrug-resistant Candida species. Curr. Opin. Infect. Dis. 2017, 30, 528–538.

[58]. Potente, G., Bonvicini, F., Gentilomi, G. A., & Antognoni, F.  Anti-Candida activity of essential oils from Lamiaceae plants from the Mediterranean area and the Middle East. Antibiotics, 2020, 395.